

# Switching between standard coral reef benthic monitoring protocols is complicated: proof of concept

Henri Vallès[1], Hazel A. Oxenford[2] and Alex Henderson[2]

[1] Department of Biological and Chemical Sciences, University of the West Indies, Cave Hill, Barbados
[2] Centre for Resource Management and Environmental Studies, University of the West Indies, Cave Hill, Barbados

## ABSTRACT

Monitoring the state of coral reefs is necessary to identify drivers of change and assess effectiveness of management actions. There are several widely-used survey methods, each of which is likely to exhibit different biases that should be quantified if the purpose is to combine datasets obtained via different survey methods. The latter is a particularly important consideration when switching methodologies in long-term monitoring programs and is highly relevant to the Caribbean today. This is because of the continuing need for regionally comparable coral reef monitoring datasets and the fact that the Global Coral Reef Monitoring Network (GCRMN)-Caribbean node is now recommending a photoquadrat (PQ) method over the chain intercept transect method widely adopted by the members of the first truly regional monitoring network, Caribbean Coastal Marine Productivity Program (CARICOMP), in the early-1990s. Barbados, a member of the CARICOMP network, has been using a variation of the chain intercept method in its long-term coral reef monitoring program for more than two decades. Now a member of GCRMN-Caribbean, Barbados is considering switching to the PQ method in conformity with other regional members. Since we expect differences between methods, this study seeks to quantify the nature of those differences to inform Barbados and others considering switching methods. In 2017, both methods were concurrently implemented at 21 permanent monitoring plots across three major reef types in Barbados. Differences in % cover estimates for the six major benthic components, that is, hard corals, sponges, gorgonians, macroalgae, turf algae and crustose coralline algae, were examined within and among types. Overall, we found a complex pattern of differences between methods that depended on the benthic component, its relative abundance, and the reef type. We conclude that most benthic components would require a different conversion procedure depending on the reef type, and we provide an example of these procedures for Barbados. The factors that likely contribute to the complex pattern of between-method differences are discussed. Overall, our findings highlight that switching methods will be complicated, but not impossible. Finally, our study fills an important gap by underscoring a promising analytical framework to guide the comparison of ecological survey methods on coral reefs.

Corresponding author
Hazel A. Oxenford,
hazel.oxenford@cavehill.uwi.edu

## INTRODUCTION

Monitoring is a fundamental part of resource management, and in the case of coral reefs is critically important for assessing their state and measuring the success of management actions (*Flower et al., 2017*; *Hill & Wilkinson, 2004*; *Rogers et al., 1994*), since coral reefs are not only highly sensitive to a wide array of natural and anthropogenic stressors (*Mumby & Steneck, 2008*), but are particularly vulnerable to the global climate crisis (*Hughes et al., 2018*). Within the Caribbean, the need for effective coral reef management interventions guided by standardized monitoring is particularly acute given that the region is experiencing widespread degradation of its reefs (*Jackson et al., 2014*) whilst also relying heavily on their ecosystem services to support tourism-dependent national economies and local livelihoods (*Burke & Maidens, 2004*; *Mumby et al., 2014*).

The use of standardized monitoring approaches is important to minimize method biases and facilitate integration of data at broad spatial and temporal scales (e.g., allowing regional comparisons and tracking of long-term changes) (*Lindeman, Kramer & Ault, 2001*). Standard biophysical methods for surveying reefs exist (*Hill & Wilkinson, 2004*; *Rogers et al., 1994*). Each method has advantages and disadvantages as well as specific biases, which has prompted a large body of work comparing them (*Beenaerts & Vanden Berghe, 2005*; *Dodge, Logan & Antonius, 1982*; *Jokiel et al., 2015*; *Leonard & Clark, 1993*; *Leujak & Ormond, 2007*; *Nadon & Stirling, 2005*; *Ohlhorst et al., 1988*; *Rogers, 1999*; *Rogers & Miller, 2001*; *Weinberg, 1981*; *Wilson, Graham & Polunin, 2007*).

Within the Caribbean, there are several variant methodologies that have been used by regional long-term monitoring programs such as the Caribbean Coastal Marine Productivity Program (CARICOMP; *Alcolado et al., 2001*), the Atlantic and Gulf Rapid Reef Assessment (*Lang et al., 2010*) and Reef Check (*Hodgson et al., 2006*). The chain intercept transect method (sensu *Rogers et al., 1994*), involving the use of a thin chain draped carefully over the substrate contour to record benthic composition in direct contact with the chain (*Ferreira, Gonççalves & Coutinho, 2001*; *Rogers, 1999*; *Rogers, Gilnack & Fitz, 1983*; *Rogers & Miller, 2001*), was adopted in the early 1990s as the standardized methodology to quantify coral reef benthic composition by the first truly regional monitoring network, CARICOMP. These data have contributed to several region-wide studies and provided the first standardized baseline across the Caribbean (*Alcolado et al., 2001*; *Cortés et al., 2019*). With the cessation of CARICOMP in 2007 and a strong recommendation from *Jackson et al. (2014)* that a standard monitoring program must be maintained in the Caribbean, there has been a re-vitalisation of the Caribbean node of the Global Coral Reef Monitoring Network (GCRMN-Caribbean) of the International Coral Reef Initiative. In their new biophysical monitoring guidelines, their recommended Level-3 (preferred) method is to use a photoquadrat (PQ) method (*GCRMN-Caribbean, 2016*).

As part of the CARICOMP network, Barbados has been using chain transects for their long-term coral reef monitoring program for more than two decades, albeit with one relevant modification. In the modified version of the chain intercept transects in Barbados, substrate composition is recorded at regularly spaced points (rather than continuously) along the chain because preliminary surveys in Barbados have indicated that point-intercept sampling along the chain yielded similar results to continuous sampling, while being less time consuming (*Allard, 1994*). This chain point-intercept (CPI) method retains a key attribute of chain intercept transects in that it captures information about the three-dimensional structure of the reef benthos (*Hill & Wilkinson, 2004*; *Rogers et al., 1994*).

Now a member of GCRMN-Caribbean, Barbados is considering switching to the PQ method in conformity with other regional members. Since we expect differences between these two methods, this study seeks to evaluate the potential magnitude and nature of method biases between the CPI methodology used in Barbados and an adapted version of the PQ level-3 methodology recommended by the GCRMN-Caribbean, with a focus on different reef types. Our main focus here is to explicitly assess whether or not PQ data conversion procedures are likely to be needed so that future PQ datasets can be meaningfully and most accurately compared with the existing CPI historical baselines. We do so using the already established and on-going long-term monitoring program of Barbados; thus broader issues pertaining to the development of an adequate coral reef long-term monitoring program (*Aronson et al., 1994*; *Brown et al., 2004*; *Green & Smith, 1997*; *Houk & Van Woesik, 2006*) are beyond this study's scope. It is expected that our findings will be relevant to the wider Caribbean community possessing historical datasets and considering this transition.

## METHODS

### Reef study sites

Barbados is located in the southeastern Caribbean and has a narrow shelf with easily accessible coral reefs all along the sheltered west and semi-sheltered southwest coasts (Fig. 1). These coral reefs are generally classified into three major types; bank, fringing and patch reefs that differ in physical location (depth, distance from shore) and structure, and in their biological community composition (*Brathwaite, Oxenford & Roach, 2018*; *Oxenford et al., 2008*; *Vallès et al., 2019*). A total of 47 reef sites spread across these reefs are surveyed every 5 years as part of the Barbados government's long-term reef monitoring program initiated in 1987 (*CERMES, 2018*). In this study 21 of these reef sites (seven of each reef type) were selected for comparing the two reef survey methodologies (Fig. 1; Table S1).

### Benthic survey methodologies

The linear CPI transect method used by the long-term Barbados Reef Monitoring Programme (BRMP) is reported in detail by *CERMES (2018)* and summarized in Table S2 and Fig. 2. In short, ten straight-line transects were surveyed at two m intervals within a 10 × 20 m permanent monitoring plot established at each reef site. A fine brass chain,
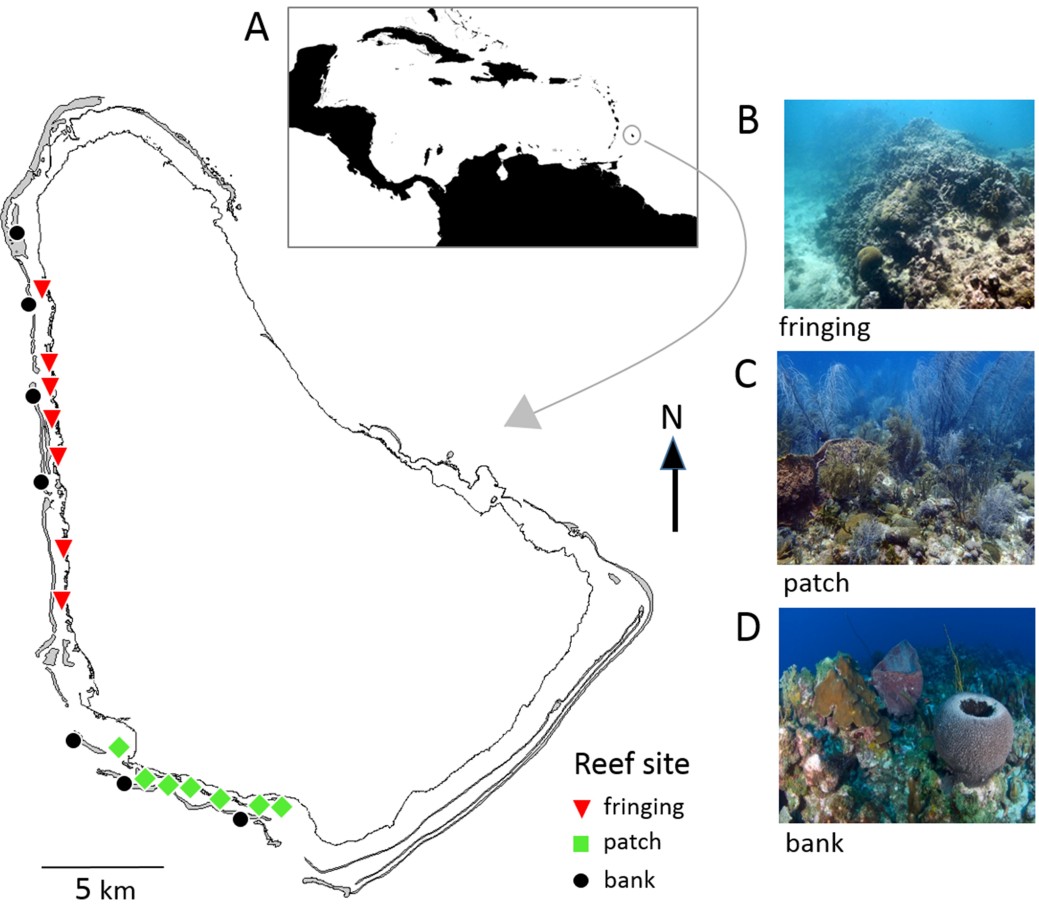

**Figure 1** **Map of Barbados showing locations of the 21 reef survey sites along the west and southwest coasts.** Inset map (A) shows position of Barbados in the southeastern Caribbean. The photographs show typical reef types in Barbados: (B) fringing reef, (C) patch reef and (D) bank reef.

marked at 10 cm intervals, was subsequently carefully laid along the reef profile under each straight-line transect. The substrate immediately under each 10 cm chain mark was identified to species level for hard corals, sponges, gorgonians and macroalgae. Turf algae and crustose coralline algae (CCA) were identified only by these broad categories to include the many species within these groups. Note that because the chains are purposely laid to follow the elevation contour of the substrate, they will extend longer than 10 m, typically yielding >1,200 point records per permanent plot (Table S2).

In this study we worked alongside the BRMP survey team from 10 July–4 Sept 2017 to undertake a PQ survey at each of 21 reef sites using a slightly modified version of the Level-3 highly recommended PQ protocol of *GCRMN-Caribbean (2016)*. The orientation, number and length of transects, and total number of PQs were slightly adapted to facilitate an appropriate comparison between the methods within the established 10 × 20 m permanent monitoring plots, whilst maintaining approximately the same number of data points per reef site for each method (Table S2). The physical setup of the monitoring plot and the two data collection protocols used at each reef site is illustrated in Fig. 2;

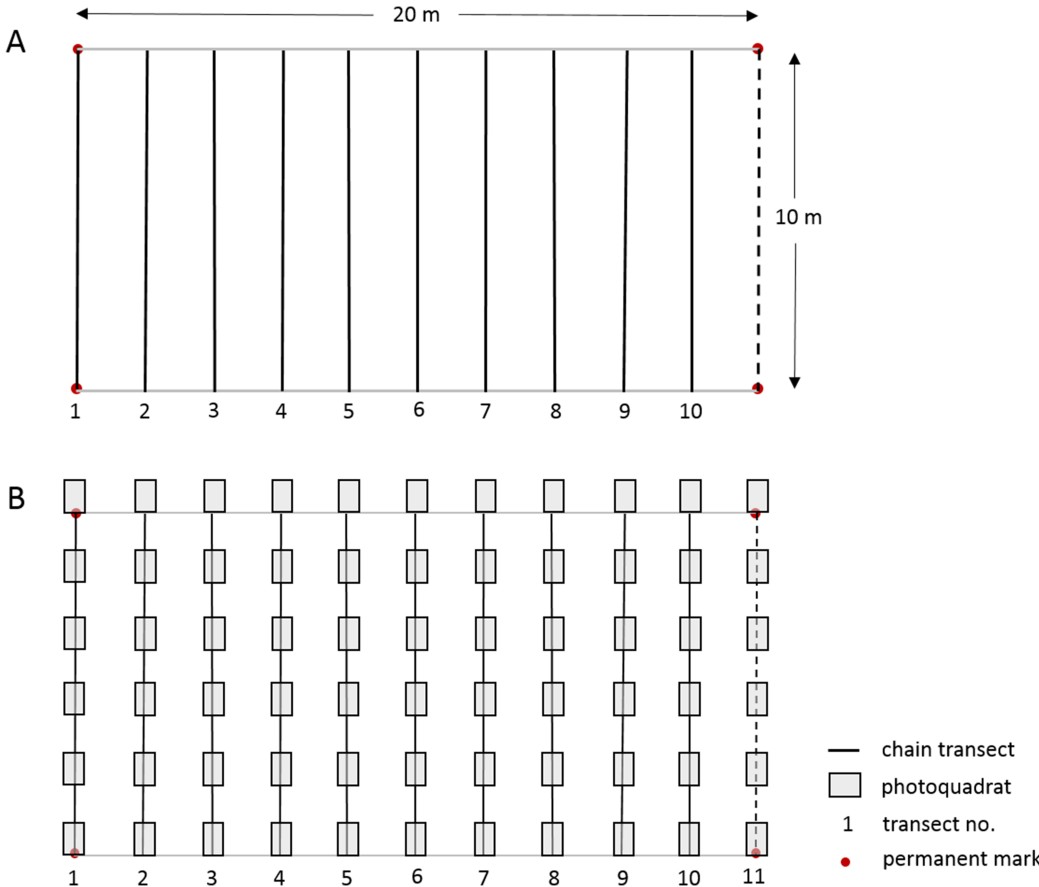

**Figure 2 Layout of 10 × 20 m permanent monitoring plot showing permanent corner marks, (A) positions of 10 temporary chain transects and (B) positions of 11 temporary transects with 66 photoquadrats.**

a more detailed comparison of the methods is summarized in Table S2. Six PQs (90 × 60 cm with a portrait orientation and spaced 110 cm apart) were taken along each of the same ten 10 m transects used for the CPI method, plus an additional 10 m transect following the permanent monitoring plot demarcation rope at the 20 m mark (Fig. 2). As such, a total of 66 PQs were taken at each monitoring site with an Olympus TG-3 camera in an external Olympus housing using the highest resolution and underwater image settings, and the internal flash for the bank reef sites to compensate for the loss of color at depth. A one m monopod was used to maintain the camera at a fixed perpendicular distance from the substrate.

## Data handling and analysis
### Raw data post-processing
For the CPI method, data were transferred from underwater slates directly to an excel database and were manipulated using the pivot table tool to produce appropriate summary tables. For the PQ method, photographs were post-processed as recommended by the GCRMN protocol using the Coral Point Count with Excel extensions (CPCe) software

(*Kohler & Gill, 2006*) to overlay 25 random points on each of the 66 images per site, yielding a total of 1,650 point records per permanent plot (Table S2). The available categories in the CPCe software were modified to match the species and category codes used in the BRMP database. Photos that were too dark due to depth, cloud cover or shadow were edited in the photo-editing software Picasa 3.9.140 (Google Inc., Mountain View, CA, USA) to increase brightness and color saturation and thereby aid in substrate identification.

### Data aggregation

A single benthic component % cover value was obtained from each permanent plot. This was achieved by simply dividing the total number of point records belonging to a given benthic component by the total number of valid point records for all components at the permanent plot.

### Assessing differences between methods in cover estimates

To assess differences between methods, we followed the approach outlined in *Altman & Bland (1983)* and *Bland & Altman (1986)*. For each site and benthic component, we subtracted the % cover estimate obtained using the CPI method from that obtained using the PQ method. We then plotted these site differences against the average % cover obtained from both methods, which we considered to be the best estimate of the true percent cover value (note that the latter is not known) (*Altman & Bland, 1983*; *Bland & Altman, 1986*). Such plotting allows for quick visual assessment of any type of scaling relationship between the difference and the average for a given benthic component at a given reef type. Such scaling relationships, if they exist, have the potential to confound differences in method biases among reef types and/or benthic components and thus need to be identified and accounted for.

In addition to plotting the differences between methods in % cover against the average % cover for both methods at each site, we also pooled these site estimates within each reef type to generate averages and 95% confidence intervals for each reef type. The 95% confidence intervals were generated via bootstrapping of site values using the "boot" package (*Canty & Ripley, 2017*) in R (*R Core Team, 2017*). Lack of a significant difference between methods would imply a random scatter of the site difference values around the zero line, which would translate into the average value for that reef type exhibiting 95% confidence intervals largely overlapping with the zero line. Similarly, the 95% confidence intervals can be used to visually compare the overall differences between methods among reef types.

### Data analysis

We examined each benthic component separately with a twofold aim: (1) to assess whether, within a given reef type, differences in % cover scaled with the average % cover obtained from both methods, and (2) to identify potential systematic differences among reef types in % cover between methods. We used an analysis of covariance (ANCOVA) framework to conduct these analyses. This involved building a linear model with % cover between methods as a response variable and reef type (categorical) and the average % cover

obtained from both methods (numerical covariate) as predictors, along with their interaction.

A statistically significant interaction between the categorical and numerical predictors would imply the existence of a scaling relationship between differences in % cover (between methods) and average % cover (of both methods) that differs in slope across reef types. This would mean that the two methods translate the *true* abundance of a given benthic component into cover estimates using functional relationships that differ fundamentally among reef types. Under such scenario, accurately converting estimates between methods within and among reef types would require prior knowledge of each of the different linear functional relationships for each reef type.

In contrast, a lack of a significant interaction term but presence of a significant effect of average % cover would imply a scaling relationship between differences in % cover and average % cover that is similar among reef types. If there is also a significant effect of reef type, then the scaling would be similar among reef types but each type would have a different baseline (intercept) value. If there is not a significant reef type term, then all the reef types would share the same baseline value. In the latter case, a single linear functional relationship would be required for accurate conversions of method estimates within and among reef types.

Finally, a lack of a significant effect of average % cover (either alone or via the interaction term) would imply the absence of any scaling relationship between differences in % cover and average % cover. This would simplify conversions between methods within reef types as it would require simply adding (or subtracting) a constant value to each estimate. If there is also a significant effect of reef type, then such a constant would differ among reef types; if not, then the constant would be the same among reef types. The latter case would be the ideal scenario because it would imply the use of a single conversion constant for all estimates, irrespective of reef type.

### Converting PQ estimates to CPI ones

Since our ultimate goal is to be able to meaningfully compare current and future PQ % cover estimates to historical CPI ones, we derived linear equations allowing the conversion of PQ estimates to CPI ones for our data. To streamline and simplify this process, we used the previous ANCOVA framework to derive the parameter estimates for the conversion equations. For each benthic component, we used the most parsimonious ANCOVA model (i.e., after removing all non-significant ($p > 0.05$) model terms) to obtain the relevant intercept and slope estimates for each reef type. Thus, for each reef type, we obtained a simple linear model linking differences in % cover between methods to average % cover as given by Eq. (1):

$$PQ - CPI = \beta 0 + \beta 1 \left( \frac{PQ + CPI}{2} \right), \tag{1}$$

where $\beta 0$ and $\beta 1$ represent the intercept and slope, respectively, of the model, and PQ and CPI represent the % cover estimates for each method.
Under the assumption that PQ and CPI cover estimates are indeed linearly related, then it follows that Eq. (1) can be re-written as Eq. (2):

$$\text{CPI} = -\frac{\beta0}{\left(1+\dfrac{\beta1}{2}\right)} + \frac{\left(1-\dfrac{\beta1}{2}\right)}{\left(1+\dfrac{\beta1}{2}\right)}\ \text{PQ}, \tag{2}$$

where the first and second fraction terms represent the intercept and slope, respectively, of the linear model allowing the direct conversion of PQ estimates into CPI ones.

We used the "gls" function in the "nlme" package (*Pinheiro et al., 2017*) in R to conduct the ANCOVA models. We allowed the variance components to differ among reef types in all models and used residual plots to assess potential violations of necessary conditions for parametric testing.

## RESULTS

Exploratory scatterplots assessing relationships between the % cover estimates of both methods revealed that both methods were strongly and significantly correlated across all sites for each benthic component (Pearson $r \geq 0.86$, $n = 21$, $p < 0.001$; Figs. 3A–3F). However, a visual inspection of the differences in % cover between methods against the average % cover of both methods revealed complex relationships that depended on both the benthic component and reef type (Figs. 4A–4F). These patterns are summarized below by benthic component.

### Hard corals

For hard corals, differences in % cover between methods did not appear to scale with the average % cover of both methods in any of the reef types (Fig. 4A). Site differences in % cover between methods were consistently negative (Fig. 4A) and, on average, appeared to differ in magnitude among some reef types, as evidenced by the lack of overlap in 95% confidence intervals between the bank and fringing reefs (Fig. 4A). The ANCOVA confirmed these results; the reef type term was statistically significant, but not the average cover covariate nor the interaction between the latter and reef type (Table 1). Thus, for hard corals, there was no evidence that differences between methods scaled significantly with average % cover. Nevertheless, the two methods differed significantly in estimates of % cover and, importantly, the extent to which they did so differed among reef types. Based on the observed average difference estimates, the PQ method underestimated cover relative to the CPI method by absolute average values of 3.2%, 5.2% and 6.2% on bank, fringing and patch reefs, respectively. This allows for a straightforward conversion to CPI by adding these absolute values to the PQ estimates (Table 2).

### Sponges

For sponges, differences in % cover between methods did not appear to scale with the average % cover of both methods in any reef type (Fig. 4B). Most site differences in % cover between methods were negative (13 out of 15 site values) (Fig. 4B), and these differences

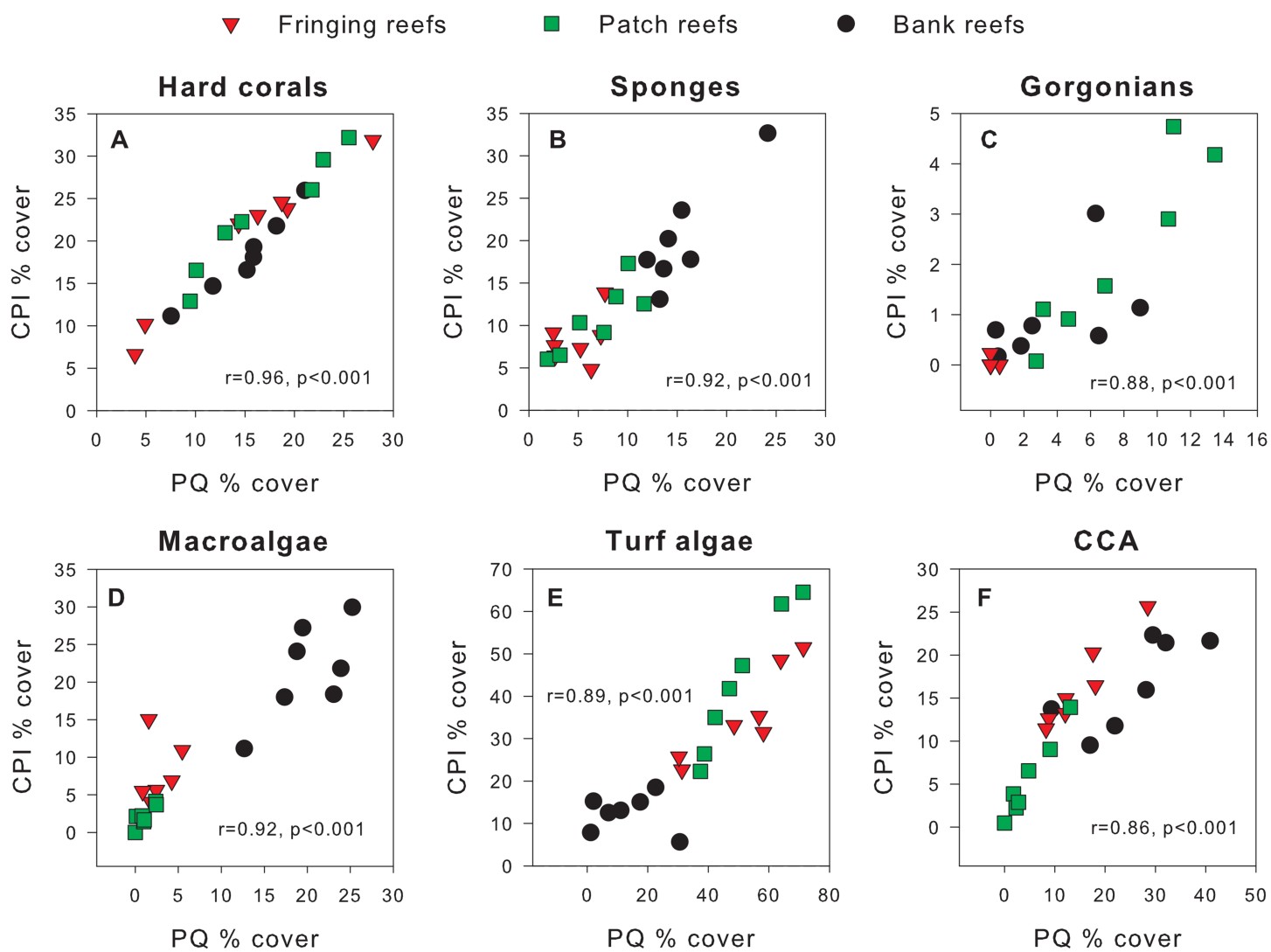

**Figure 3 Scatterplots showing original percent cover values obtained using the chain point-intercept (CPI) method vs. the photoquadrat (PQ) method for each benthic component on the three reef types.** Benthic components shown are (A) hard corals, (B) sponges, (C) gorgonians, (D) macroalgae, (E) turf algae and (F) crustose coralline algae. Pearson correlation coefficients are also shown along with their corresponding *p*-values (*n* = 21).

did not appear to differ in magnitude among reef types, as evidenced by the overlap in 95% confidence intervals between all reef types (Fig. 4B). This was confirmed by the ANCOVA, which indicated that the reef type term was not statistically significant, nor was the average cover covariate or the interaction between the latter and reef type (Table 1). Thus, for sponges, the two methods did not differ significantly in estimates of % cover among reef types. However, the intercept term of the ANCOVA was significant (Table 1) indicating that the two methods still differed in their overall estimates. Pooling the observed estimates across reef types indicated that the PQ method underestimated cover relative to the CPI method by an absolute average value of 3.9% irrespective of reef type. Thus, converting PQ % cover estimates to CPI would require adding this absolute value to the PQ estimates (Table 2).

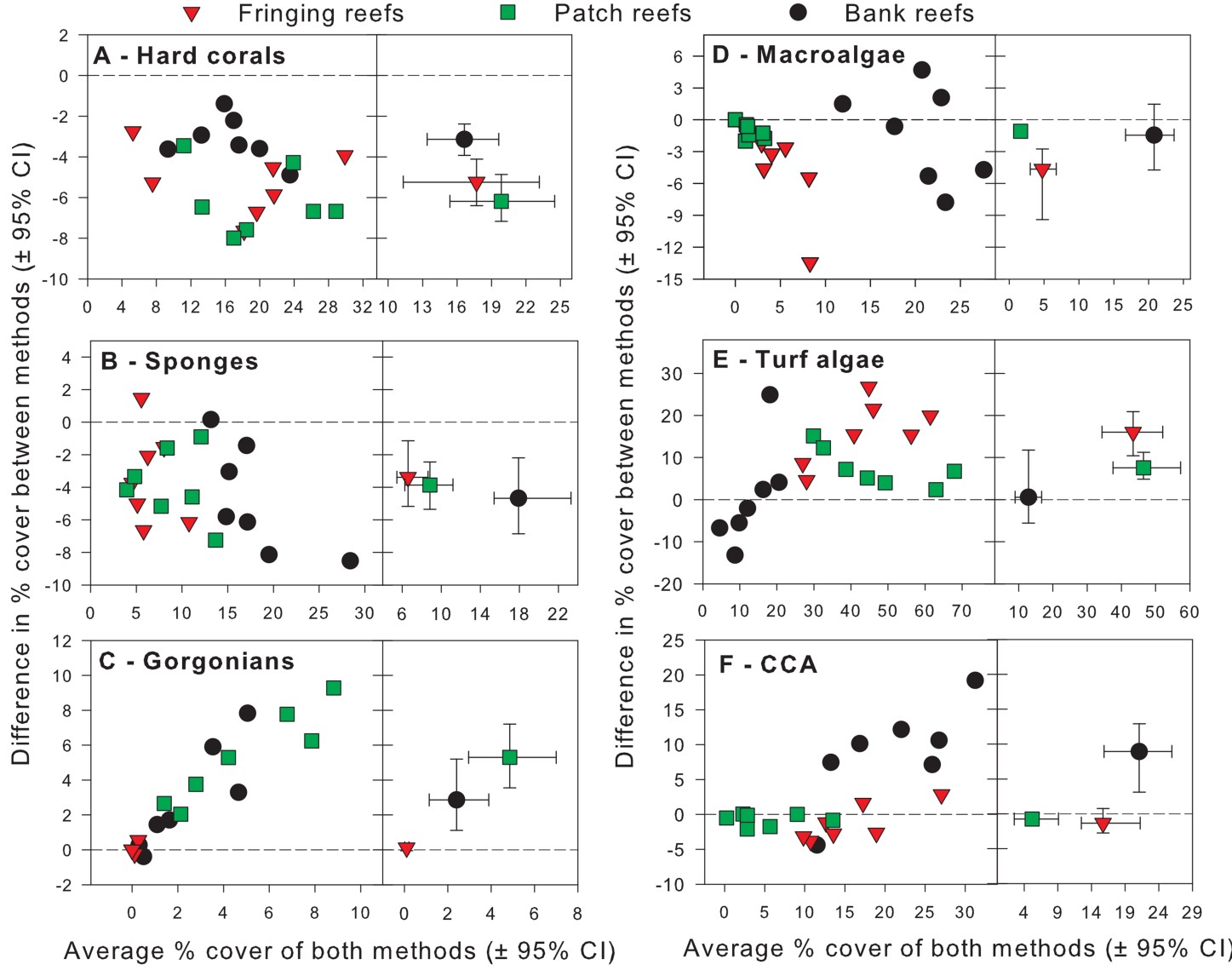

**Figure 4 Differences in percent cover estimates between methods against the average percent cover estimated by both methods for each benthic component across the three reef types in Barbados.** Benthic components shown are (A) hard corals, (B) sponges, (C) gorgonians, (D) macrolagae, (E) turf algae and (F) crustose coralline algae. Each panel shows site-specific values (left) and averages for each reef type (right) with corresponding 95% bootstrap confidence intervals ($n = 7$).

## Gorgonians

For gorgonians, differences in % cover between methods strongly scaled with the average % cover of both methods on the patch and bank reefs, and they did so in a manner that appeared consistent between these two reef types (Fig. 4C). In contrast, site differences on the fringing reefs, which showed very low values for the averages between methods, revolved closely around the zero line (Fig. 4C). On average, and taken at face value, differences between methods scaled with average % cover across reef types and the lack of overlap in 95% confidence intervals supported that reef differences were statistically significant (Fig. 4C). The ANCOVA showed a highly significant effect of both the reef type

**Table 1 Summary of the ANCOVA full model results for each benthic component, comparing difference in percent cover as a function of reef type and average percent cover.** Models use the differences in % cover between methods as a response variable, and reef type (categorical) and the average % cover obtained from both methods (numerical) as predictors, along with their interaction. Bold font indicates significant terms ($p < 0.05$). See Table 2 for the interpretation of these findings.

| Benthic component | Term | d.f. num | d.f. den | F statistic | p-value |
|---|---|---|---|---|---|
| Hard corals | Intercept | 1 | 15 | 180.91 | **<0.001** |
| | Reef type | 2 | 15 | 8.36 | **0.004** |
| | Average cover | 1 | 15 | 1.22 | 0.288 |
| | Reef type × Average cover | 2 | 15 | 0.09 | 0.915 |
| Sponges | Intercept | 1 | 15 | 50.02 | **<0.001** |
| | Reef type | 2 | 15 | 0.41 | 0.673 |
| | Average cover | 1 | 15 | 4.27 | 0.057 |
| | Reef type × Average cover | 2 | 15 | 0.58 | 0.573 |
| Gorgonians | Intercept | 1 | 15 | 23.97 | **<0.001** |
| | Reef type | 2 | 15 | 101.00 | **<0.001** |
| | Average cover | 1 | 15 | 62.48 | **<0.001** |
| | Reef type × Average cover | 2 | 15 | 2.45 | 0.120 |
| Macroalgae | Intercept | 1 | 15 | 27.61 | **<0.001** |
| | Reef type | 2 | 15 | 4.74 | **0.025** |
| | Average cover | 1 | 15 | 8.37 | **0.011** |
| | Reef type × Average cover | 2 | 15 | 1.21 | 0.324 |
| Turf algae | Intercept | 1 | 15 | 65.55 | **<0.001** |
| | Reef type | 2 | 15 | 7.89 | **0.005** |
| | Average cover | 1 | 15 | 1.15 | 0.301 |
| | Reef type × Average cover | 2 | 15 | 7.07 | **0.007** |
| CCA | Intercept | 1 | 15 | 3.93 | **0.066** |
| | Reef type | 2 | 15 | 13.31 | **<0.001** |
| | Average cover | 1 | 15 | 5.34 | **0.035** |
| | Reef type × Average cover | 2 | 15 | 5.23 | **0.019** |

Note:
CCA, crustose coralline algae.

and the gorgonian average abundance covariate (Table 1). However, the interaction term was not significant, indicating a consistent scaling relationship across reef types (Table 1). Thus, for gorgonians, differences between methods scaled significantly (and positively) with average % cover in a manner that was consistent among reef types. Nevertheless, the baseline values (intercepts) differed significantly among reef types. In summary, accurately converting PQ % cover values to CPI would require a linear transformation with the same slope but different baseline values among reef types (Table 2). However, assessment of the residual plots for this benthic group indicated some evidence of heterogeneity of variance (despite the use of different variance components for each reef type). This warrants extra caution in the interpretation of the significance of some of the model terms, although it is unlikely that this would affect the visually obvious scaling relationship with average % cover (Fig. 4C).

**Table 2 Summary of conclusions from the comparisons of percent cover estimates between chain point-intercept (CPI) and photoquadrat (PQ) methods for six benthic components on the three reef types in Barbados.** The corresponding formulae to convert percent cover between methods is shown for each benthic component at each reef type (last two columns). Benthic components are ordered (from top to bottom) by increasing complexity in the pattern of differences between methods.

| Benthic component | Conclusion | Reef type | Conversion formulae |
|---|---|---|---|
| Sponges | Systematic differences in % cover between methods that do not depend on reef type and which do not scale with % cover | Fringing | $CPI = 3.90 + PQ$ |
| | | Patch | $CPI = 3.90 + PQ$ |
| | | Bank | $CPI = 3.90 + PQ$ |
| Hard corals | Systematic differences in % cover between methods that depend on reef type, but which do not scale with % cover | Fringing | $CPI = 5.24 + PQ$ |
| | | Patch | $CPI = 6.16 + PQ$ |
| | | Bank | $CPI = 3.15 + PQ$ |
| Macroalgae | Systematic differences in % cover between methods that depend on reef type and which scale with % cover in a manner that is consistent among reef types | Fringing | $CPI = 3.07 + 1.65 \times PQ$ |
| | | Patch | $CPI = 0.37 + 1.65 \times PQ$ |
| | | Bank | $CPI = -11.48 + 1.65 \times PQ$ |
| Gorgonians | Systematic differences in % cover between methods that depend on reef type and which scale with % cover in a manner that is consistent among reef types | Fringing | $CPI = -0.03 + 0.36 \times PQ$ |
| | | Patch | $CPI = -0.45 + 0.36 \times PQ$ |
| | | Bank | $CPI = -0.39 + 0.36 \times PQ$ |
| CCA | Systematic differences in % cover between methods that depend on reef type and which scale with % cover in a manner that differs among reef types | Fringing | $CPI = 5.71 + 0.71 \times PQ$ |
| | | Patch | $CPI = 0.74 + 1.00 \times PQ$ |
| | | Bank | $CPI = 4.81 + 0.46 \times PQ$ |
| Turf algae | Systematic differences in % cover between methods that depend on reef type and which scale with % cover in a manner that differs among reef types | Fringing | $CPI = 0.62 + 0.67 \times PQ$ |
| | | Patch | $CPI = -20.80 + 1.26 \times PQ$ |
| | | Bank | $CPI = 10.84 + 0.13 \times PQ$ |

**Note:**
CCA, crustose coralline algae.

## Macroalgae

For macroalgae, differences in % cover between methods appeared to scale negatively with the average % cover of both methods in most reef types (Fig. 4D). Most site differences in % cover between methods were negative (12 out 15 site values) (Fig. 4D). On average, and taken at face value, these differences appeared to differ in magnitude among some reef types, as evidenced by the lack of overlap in 95% confidence intervals between patch and fringing reefs (Fig. 4D). The ANCOVA confirmed the significant effect of both reef type and the average % cover covariate, but failed to find a significant effect of the interaction between the two (Table 1). Thus, for macroalgae, differences between methods scaled significantly (and negatively) with average % cover in a manner that was consistent among reef types. Nevertheless, the baseline values (intercepts) differed significantly among reef types. Thus, accurately converting PQ % cover to CPI would require a linear transformation with the same slope but different baseline values among reef types (Table 2).

## Turf algae

For turf algae, differences in % cover between methods scaled strongly with the average % cover of both methods in a way that differed among reef types (Fig. 4E). For example, whereas site differences in % cover between methods appeared to increase with average

cover on both fringing and bank reefs, they appeared to decrease on patch reefs (Fig. 4E). On average, and taken at face value, differences between methods appeared to differ in magnitude among reef types, as evidenced by the minimum overlap in 95% confidence intervals between the patch and fringing reefs (Fig. 4E). The ANCOVA confirmed the presence of a highly significant interaction term between average % cover and reef type; reef type (but not the average cover covariate) was also statistically significant (Table 1). Thus, the magnitude of the differences between methods scaled significantly with average % cover in a way that differed among reef types, precluding straightforward overall comparisons within and among reef types. In summary, accurately converting PQ % cover to CPI would require a linear transformation with different slopes and different baseline values among reef types (Table 2).

### Crustose coralline algae

For CCA, differences in % cover between methods scaled with the average % cover of both methods in some reef types (e.g., bank reefs) but not others (e.g., patch reefs) (Fig. 4F). Moreover, site differences in % cover between methods were relatively small in patch and fringing reefs, but large (and mainly positive) on the bank reefs (Fig. 4F). On average, and taken at face value, these differences appeared to differ in magnitude among reef types, as evidenced by the lack of overlap in 95% confidence intervals between the bank reefs and the other two reef types (Fig. 4F). The ANCOVA revealed that the interaction term between average % cover and reef type was significant, as were all the other terms in the model (Table 1). Thus, the magnitude of the differences between methods scaled significantly with average % cover in a way that differed among reef types, precluding straightforward comparisons within and among reef types. In summary, accurately converting PQ % cover to CPI would require a linear transformation with different slopes and different baseline values among reef types (Table 2).

Table 2 provides a summary of our main findings for each benthic component and the linear functions to convert PQ % cover estimates to CPI ones for each reef type using Eq. (2) (see Figs. S1A–S1F for a graphic display of these conversions for each benthic component). Replacing the original PQ values with these converted ones removes all visual evidence of systematic biases and scaling relationships between methods, as expected (see Figs. S2A–S2F).

## DISCUSSION

We have demonstrated statistically significant differences in estimates of % cover between the PQ and CPI method at our reefs for the most commonly used benthic components in coral reef monitoring programs. Importantly, we have shown that the magnitude and nature of such differences depends variously on the benthic component of interest, the abundance (% cover) of that component, and the type of reef examined.

Beyond showing that differences between methods are complex and statistically significant, we further argue that the magnitude of these method biases is important from a management perspective and should not be ignored. We illustrate this point by using additional CPI data from the BRMP to assess changes in hard coral % cover between 2012

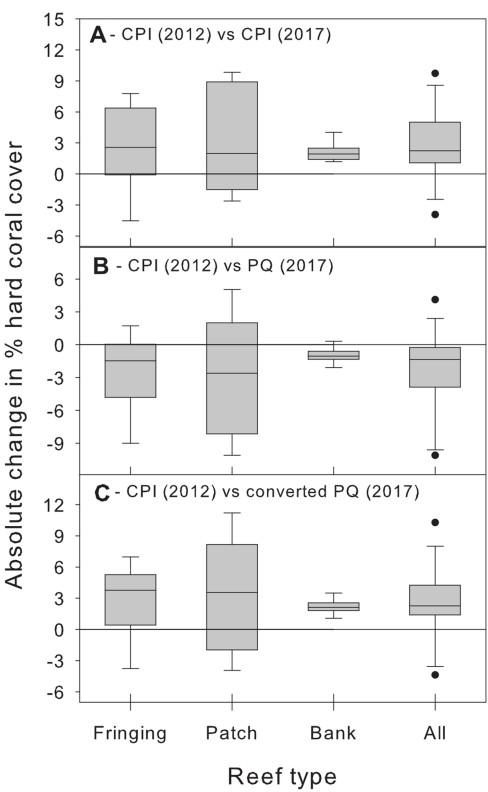

**Figure 5 Changes in absolute hard coral percent cover between 2012 and 2017 estimated by comparing data from 2012 obtained using the chain point-intercept (CPI) method with method-specific data sets from 2017.** Comparisons are shown between 2012 CPI data and (A) 2017 CPI data, (B) 2017 PQ data and (C) 2017 PQ data converted to CPI. Data are for the 21 reef survey sites. The horizontal reference line indicates zero change. A median test (after combining data from all reef types) confirms that the median change between 2012 and 2017 is statistically significant (i.e., different from zero) in all three temporal comparisons (all cases: $W \geq 47$, $n = 21$, $p \leq 0.017$).

and 2017 at the same 21 study sites. Comparing the 2012 estimates with those of 2017 obtained using the same CPI methodology reveals a statistically significant (albeit small) overall increase in absolute % cover across all reef types (Fig. 5A). In contrast, comparing the 2012 CPI estimates with the original 2017 PQ ones reveals the opposite pattern, that is, a small yet statistically significant overall decrease in hard coral absolute % cover (Fig. 5B). The true trend is recovered once the PQ data are converted to CPI estimates (Fig. 5C). Clearly, switching to PQ data without adequately accounting for method differences would have obscured this modest, yet optimistic, signal and led to an erroneous conclusion about recent trends in coral cover in Barbados by masking the upward trend in coral cover and suggesting a decline instead.

Thus, our study shows that (1) % cover estimates obtained using the PQ method will need to be converted to allow for meaningful comparisons with historical data obtained using the CPI method in Barbados and (2) the nature of this conversion will differ depending on the benthic component of interest and the type of reef examined (Table 2).

What drives this complex pattern of differences between methods in cover estimates? Following *Leujak & Ormond (2007)*, we consider a number of potentially interacting factors, namely (1) the contour effect, (2) the proportion of substrate sampled, (3) the angle of view, (4) image resolution, (5) observer bias and (6) data calculation.

The contour effect results from the fact that the PQ method samples the benthos through a 2D horizontal plane (bird's-eye view) and as such it will systematically ignore information about the vertical contour (rugosity) of the benthos, whereas the CPI method purposely samples along that vertical contour (*Hill & Wilkinson, 2004*; *Rogers et al., 1994*). This implies that benthic features exhibiting a pronounced vertical dimension will likely be underestimated by the PQ relative to the CPI method. This might explain why the PQ method yielded consistently lower cover estimates for hard corals and sponges than the CPI method across all reef types in our study, as these are the benthic components typically responsible for most of the vertical structure on coral reefs. Interestingly, and in contrast to our findings, *Rogers (1999)* and *Rogers & Miller (2001)*, who compared video images with the chain intercept transect method in what are likely the most comparable studies to ours, found no differences in coral cover between methods. Since objects close to the camera will appear larger than those of the same size but further away from the camera (*Porter & Meier, 1992*), this might have partially counterbalanced the contour effect in their studies, suggesting that the overall physical structure of the reef, as well as the height of the camera above the substrate (e.g., only ~40 cm in Roger & Miller's study vs. ~100 cm in ours), are important. Furthermore, the fact that differences between methods in coral cover estimates varied in magnitude across reef types is not surprising. Several authors have reported site-specific differences in method biases, attributing these to differences in spatial heterogeneity (*Dodge, Logan & Antonius, 1982*), physical complexity (*Nadon & Stirling, 2005*) or coral cover (*Lam et al., 2006*).

Another important factor is the considerable difference between methods in the proportion of benthic substrate that was effectively sampled within the permanent plot. Although we ensured that the number of sample points used within each plot was roughly similar between methods, the effective area sampled differed by one order of magnitude. The PQ method employed here involved sampling points distributed across 66 0.54-$m^2$ quadrats, representing a total sampled area of 35.6 $m^2$ (17.8%) out of the 200 $m^2$. In contrast, the CPI method involved sampling points distributed along ten chains averaging ~13 m in length (depending on the reef contour) and just 0.005 m wide, representing a total sampled area of little more than 0.65 $m^2$ (0.33%). This is an intrinsic limitation of all line transect methods and the consequence is that, everything else being equal, they are more likely to underestimate the least abundant benthic components (*Leujak & Ormond, 2007*). Thus, the PQ method should provide more representative estimates of benthic cover at low abundance, particularly in the inherently heterogeneous benthos of most coral reefs. In our study, such an effect should have translated into higher PQ estimates when average abundance (% cover) of the benthic component was low, if the rarer benthic components were systematically missed by the CPI method. Yet, this was generally not the case (Figs. 4A–4F), suggesting that the difference in effective area sampled was not an important factor. The pooling of data into the six broad benthic

components might have contributed to reduce its influence, but this factor might become important when individual taxa are examined separately (*Leujak & Ormond, 2007*) or if some of the broad benthic components (e.g., hard corals which currently average 20.4% cover across the study sites) were to decrease considerably in overall abundance in the future.

*Leujak & Ormond (2007)* also highlight variability in the angle of view (parallax error) as a potentially important factor leading to differences in estimates between methods. This factor was minimized in the PQ method by using a monopod, which was consistently placed perpendicular to the main plane of the substrate. In contrast, this factor could have been particularly problematic if the chain used by the CPI method to sample the benthic component had been maintained taut between the transect ends with minimum physical contact with the substrate itself. Under such conditions, small changes in the observer's angle of view would have affected what benthic component was perceived to be directly perpendicular to the sampling chain; as a consequence, the most dominant benthic components would have likely been overrepresented. This effect would be further exacerbated under sea conditions (swells or currents) that can shift the position of the suspended line. However, because the CPI method uses a non-buoyant chain that is draped along the substrate contour (rather than kept taut away from the substrate), viewing angle artifacts are likely to have been small (but not impossible, particularly in instances where the chain overhangs vertically, or under strong swells) compared to line transect methods where the overhanging line is kept taut (*Dodge, Logan & Antonius, 1982*). Furthermore, because multiple chains were laid within the same permanent plot, the potential effects of parallax and chain shifts would have been more likely to be averaged out at the scale of the entire plot. Thus, overall, we suspect that this factor was not highly influential in our study.

Image resolution of underwater digital cameras has been a major constraint in the development of PQ and video methods for benthic coral reef monitoring (*Brown et al., 2004*; *Carleton & Done, 1995*; *Lam et al., 2006*; *Ninio et al., 2003*; *Preskitt, Vroom & Smith, 2004*). However, photographic technology is improving rapidly and so we were able to consistently obtain image resolution of appropriately high quality for most benthic groups throughout our study. For example, our identification of hard coral species using PQ, agreed closely with our in-situ coral identification using the CPI method (*Henderson, 2017*), supporting high accuracy in distinguishing between hard coral and other benthic components. However, this was not always the case for some small sponges and macroalgae patches captured by the CPI, which might have been occasionally misidentified as other benthic components in the PQ method due to insufficient resolution.

Overall, the biggest PQ image processing challenge we faced was distinguishing between a mixture of CCA, macroalgae and sometimes turf algae on the bank reefs because of the dimmer images that result from greater depths, a common problem in these types of studies (*Rogers, 1999*; *Rogers & Miller, 2001*). It is therefore possible that the consistently higher CCA values obtained using the PQ method on the bank reefs relative to the CPI method might partially reflect an incorrect systematic attribution of sampling points to CCA in lieu of macroalgae and/or turf algae cover. In spite of this, we found our PQ

method to be generally satisfactory in the context of the broad benthic components that were of interest in this study, in line with other studies (*Aronson et al., 1994*; *Ninio et al., 2003*). Obviously, the level of image resolution needed will be dictated by the level of taxonomic resolution of interest and in that regard we found that identifying some small macroalgae and small sponges to species level was often difficult. Likewise, the flexibility of using multiple cues for in-situ identification offered by the CPI method cannot be sufficiently overstated. This issue should be carefully borne in mind if one of the ultimate purposes of the PQ method is to provide photographic records of the benthic communities for potential use in ways that go beyond simple monitoring. Supplementing PQ with in situ notes for the rarer and more cryptic taxa seems a promising approach to circumvent some of the identification problems (*Preskitt, Vroom & Smith, 2004*).

Although we recognize the potentially important role that observer bias can play when collecting benthic data, we believe that this was not an important factor in generating differences between methods in our study. Using a chain draped over the substrate minimized any potential bias associated with identifying the substrate directly beneath the chain, which is a problem when the line is taut and suspended above the substrate (*Leujak & Ormond, 2007*). Taxonomic identification of reef benthic organisms can be quite challenging, particularly when these are small-sized and rare (*Ohlhorst et al., 1988*). However, the use of broad benthic categories by our study meant that most of the sessile organisms recorded in situ (CPI) and with sufficient resolution in the images (PQ) could be allocated to the appropriate category by any observer after minimum training. This is in line with inter-observer variability assessments of taxa identification in the field (*Nadon & Stirling, 2005*) and from photographic images (*Ninio et al., 2003*).

How the % benthic cover estimates are derived from the raw data is another factor potentially leading to differences in estimates between methods. In both methods, calculation of % benthic cover involved dividing the total number of records belonging to a given benthic component by the total number of records for all components within the permanent plot. This necessarily implies some degree of dependency between cover estimates of the different benthic components since their sum should be equal to 100%. By extension, summing the % cover differences between methods across all benthic components should yield a value of 0%. This implies that large negative differences between methods for some benthic components will necessarily artificially lead to large positive differences for other benthic components, and thus to an apparent over-representation for the latter. Thus, at the reef type level, the consistently higher estimates of turf algae on the fringing and patch reefs and of CCA on the bank reefs obtained by the PQ method (Figs. 4E–4F) are, to some degree, a likely artifact of the consistently lower estimates for corals and sponges, obtained by this method (Figs. 4A and 4B). In any case, this point highlights the problem of interpretation of % cover values particularly when data are obtained using planar view approaches as these cannot track associations between the substrate contour and the different benthic components. Thus, although we agree with *Leujak & Ormond (2007)* assertion that two-dimensional planar projections as derived by photographic methods provide a standardized measure of

substrate cover that is independent of reef physical complexity, the latter ceases to be the case once the raw data are transformed into percentages, which is the general rule.

Beyond *Leujak & Ormond (2007)* six aforementioned factors, we also detected a very strong "gorgonian effect" whereby differences between methods in gorgonian cover estimates increased in magnitude with overall gorgonian abundance. This effect resulted from the two methods interacting quite differently with the morphological features of gorgonians. The CPI method sampled the relatively small horizontal surface area of the gorgonian holdfasts. In contrast, the photographs of the PQ method were often dominated by the larger fan- and tree-like vertical bodies of the individual gorgonians, in spite of the camera's top–down view, ultimately yielding PQ estimates that scaled with gorgonian abundance at a greater rate than in the CPI method (Fig. 4C). This phenomenon was often exacerbated by currents and swells pushing gorgonians into the frame of the photo and adding an additional source of method bias. For the same reasons, *Rogers (1999)* and *Rogers & Miller (2001)* also found that estimates of gorgonian cover using video were consistently higher than those obtained using a chain method. Furthermore, revisiting their data shows that differences between methods also scaled with average gorgonian abundance, suggesting that this is a common phenomenon, although this specific aspect was not explicitly investigated by these authors. Furthermore, unlike CCA and turf algae, the scaling relationship for gorgonians was similar among reef types, which likely reflects an overriding effect of the interaction between the peculiar gorgonian morphology and their inclusion in the photo images. Having said that, this peculiar morphology implies that their abundance is likely better assessed using approaches that do not rely on % benthic cover estimates such as colony counts on belt transects (*Rogers et al., 1994*). However, where gorgonians are quite abundant and the PQ method is still used to assess the other benthic components, field and lab protocols should be put in place to minimize gorgonian overrepresentation on the PQ images.

We detected statistically significant differences between methods in cover estimates for all benthic components examined. We believe the analytical framework used here (*Altman & Bland, 1983*; *Bland & Altman, 1986*) has allowed for a more sensitive and in-depth assessment of the differences between methods. For example, coral reef studies comparing cover estimates between methods tend to use statistical approaches that would have ignored the scaling relationships identified in our study for most benthic components (e.g., ANOVAs and *t*-tests and their non-parametric analogs) (*Beenaerts & Vanden Berghe, 2005*; *Carleton & Done, 1995*; *Dodge, Logan & Antonius, 1982*; *Jokiel et al., 2015*; *Weinberg, 1981*). When such scaling relationships exist, not accounting for them will result in higher unexplained variance and consequently in lower power to detect method-specific effects. In that line, a lack of significant differences between methods is often interpreted as methods being comparable, yet few studies actually address the issue of statistical power during such comparisons (but see *Carleton & Done, 1995*; *Long et al., 2004*). Data transformation might help in some cases but such approach misses out on an opportunity to better understand what drives these differences. We also concur with *Bland & Altman (1986)* in that the use of correlations between method estimates (*Bouchon, 1981*; *Wilson, Graham & Polunin, 2007*) is neither technically suitable nor informative enough if the

specific goal is to assess whether data from different methods can be used interchangeably. The latter is best exemplified by *Beenaerts & Vanden Berghe (2005)*, who found % coral cover estimates between methods to be highly correlated, in line with our own findings (Figs. 3A–3F), yet their methods also significantly differed in their absolute estimates of % coral cover. Clearer statistical and biological criteria are therefore needed to guide such method comparisons.

Finally, to streamline and simplify the process of generating functions to convert PQ % cover estimates to CPI ones, we have here made use of the already existing ANCOVA statistical framework and outputs to (indirectly) derive the parameter estimates for the conversion functions (Eq. (2)). An alternative to this approach is to conduct major axis (MA) regression (a type of Model II regression) (*Legendre & Legendre, 2012*) using CPI and PQ directly as response and independent variables, respectively, for each reef type where relevant. Under a MA regression framework, both variables (CPI and PQ) are assumed to be subject to similar random sampling errors, which in this case is a reasonable expectation; MA regression will yield slopes that are symmetrical between methods (*Legendre & Legendre, 2012*). In contrast, the ANCOVA framework assumes that the independent variable (i.e., average % cover between methods) is measured with negligible sampling error, an unlikely scenario in many biological studies (*Quinn & Keough, 2002*) and in our specific context. Although this will not affect conclusions about the existence of statistically significant scaling relationships, it might ultimately lead to somewhat biased slope parameters (*Quinn & Keough, 2002*). On the other hand, the ANCOVA framework has the practical advantage of readily allowing for the integration of different variance components and for data pooling across reef types, when appropriate. We are not aware of any publicly available software that integrates both approaches. Since both approaches make different assumptions about sampling error and use the available data in different forms, they will generally result in different slope parameter estimates (and by extension, in different intercepts). In our case, and for those benthic components that exhibited scaling relationships, which approach is ultimately used to subsequently derive the model parameters made little difference to our converted PQ % cover estimates (Table S3). However, how these different approaches perform on datasets independently obtained from the ones that were used to develop the conversion models (i.e., model validation) still remains to be seen and highlights an important area for future research that could include both empirical and simulation studies.

## CONCLUSION

In conclusion, any attempt to transition between chain intercept transect methods and PQ methods will require describing the nature of the differences between methods, which is very likely to depend on both the benthic components of interest as well as the reef type. We suspect that this conclusion also applies to comparisons involving benthic methods other than the ones specifically examined in this study. It is recommended that the transition period involves surveying as many permanent sites (or sites of special interest) as possible using both methods so as to quantify as accurately as possible the varying nature of those relationships. Indeed, in this study we only had seven sites per reef type and

it is possible that some of our analyses might have suffered from low power to detect more subtle differences. Importantly, future work should examine in more detail the practical, empirical and theoretical merits of different between-method conversion procedures, which was beyond the scope of our study. Finally, we believe that an analytical framework like the one presented in this paper can guide and help standardize the process of comparing reef survey methods, even beyond the context of benthic monitoring programs.

## ACKNOWLEDGEMENTS

We acknowledge the Coastal Zone Management Unit of the Government of Barbados for use of the 2012 and 2017 CPI data from the Barbados Reef Monitoring Programme (BRMP). We also acknowledge the significant contributions of survey divers Renata Goodridge, Amy Cox, Annabel Cox and Julian Walcott; and data processing and management by Annabel Cox and Amy Cox.

### Funding

This work was supported by a postgraduate grant to Alex Henderson from the Centre for Resource Management and Environmental Studies, University of the West Indies.
The funders had no role in study design, data collection and analysis, decision to publish, or preparation of the manuscript.

### Grant Disclosures

The following grant information was disclosed by the authors:
Centre for Resource Management and Environmental Studies, University of the West Indies.

### Competing Interests

The authors declare that they have no competing interests.

### Author Contributions

- Henri Vallès conceived and designed the experiments, analyzed the data, prepared figures and/or tables, authored or reviewed drafts of the paper, approved the final draft.
- Hazel A. Oxenford conceived and designed the experiments, performed the experiments, contributed reagents/materials/analysis tools, prepared figures and/or tables, authored or reviewed drafts of the paper, approved the final draft.
- Alex Henderson performed the experiments, authored or reviewed drafts of the paper, approved the final draft.

### Data Availability

Raw data is available in the Supplemental Files.
## Supplemental Information

Supplemental information for this article can be found online at http://dx.doi.org/10.7717/peerj.8167#supplemental-information.

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
