# Peer review of "Switching between standard coral reef benthic monitoring protocols is complicated: proof of concept"

_PeerJ, doi:10.7717/peerj.8167_

## Round 0.1 · original submission · Minor Revisions

Despite the delays caused by the summer recess we finally have three excellent reviews of your paper, all of which agree that it requires only minor revisions. I have few comments to add other than, 1) I have see "continuous sampling" referred to as Line Intercept Transects (LIT) and 2) I was very interested in your observation of how the vertical dimension of benthos impacts on their assessment. I wondered if you had enough data on different generalised morphologies to demonstrate this with your data? For example we might expect to see consistent differences between boulder corals and branching or between encrusting and erect sponge morphologies?

·

Basic reporting

This paper provides a clear and detailed comparison of two established methods for assessing coral reefs. It provides an excellent example of the pit-falls associated with changing a method halfway through a monitoring programme and the work required to fully understand the magnitude of these changes.

The readability of the abstract could be improved, for example the fourth sentence of the abstract is very long. Otherwise, the writing style is very good.

Experimental design

The paper presents the information in a clear format and the statistical analyses appear appropriate for the objectives of the study. The discussion adequately explains why discrepancies have appeared between the two coral methods.

How did the authors select the number of random points to use in CPCe? A rationale for the use of 25 points would be helpful. This is touched upon in the discussion from line 361 onwards. I think a clearer statement in the methods explaining the aligned of points between methods would be useful at that point. Although 25 points have been selected here, will the monitoring also use 25 points (which doesn’t seem many to me)? If not, what are additional implications of increasing this number on detection etc?

All aspects of the abundance (via cover) has been covered in great detail by following the Altman & Bland / Bland & Altman comparison method. However, I’m left wondering about the detection of species and the reporting of site richness. It would be interesting to see a table summarising the richness at each reef type according to both methods. I suspect richness is not a monitoring requirement (as you have only used broad Reef Check groupings in the paper) but is of general interest (especially as you say most analysis is done to species level).

An estimate on the additional processing time required for PQ would also be interesting when compared with the chain method.

Validity of the findings

The specific change in methods is perhaps only an issue in one country (and consequently one specific set of reef morphologies / species assemblages) which limits its interest elsewhere. It might help if the authors look for other countries also having to make similar changes. However, I like the analysis and how it has been related to specific environmental and biological aspects.

I would have liked a little more discussion on whether this move to PQ fundamentally supports the objectives of the monitoring (i.e. was the change sufficiently informative to jeopardise a time-series…which I assume it was). Also, the relative pros and cons of both methods would be interesting, i.e. replication vs processing time, detection of rare events (e.g. disease and human litter), transparency etc. I appreciate that much of this is covered in other publications but a brief summary relating to local considerations and objectives would be interesting.

Additional comments

This paper provides a clear and detailed comparison of two established methods for assessing coral reefs. It provides an excellent example of the pit-falls associated with changing a method halfway through a monitoring programme and the work required to fully understand the magnitude of these changes. The specific change in methods is perhaps only an issue in one country (and consequently one specific set of reef morphologies / species assemblages) which limits its interest elsewhere. However, I like the analysis and how it has been related to specific environmental and biological aspects.
The paper presents the information in a clear format and the statistical analyses appear appropriate for the objectives of the study. The discussion adequately explains why discrepancies have appeared between the two coral methods.
The readability of the abstract could be improved, for example the fourth sentence of the abstract is very long. Otherwise, the writing style is very good.
How did the authors select the number of random points to use in CPCe? A rationale for the use of 25 points would be helpful. This is touched upon in the discussion from line 361 onwards. I think a clearer statement in the methods explaining the aligned of points between methods would be useful at that point. Although 25 points have been selected here, will the monitoring also use 25 points (which doesn’t seem many to me)? If not, what are additional implications of increasing this number on detection etc?
All aspects of the abundance (via cover) has been covered in great detail by following the Altman & Bland / Bland & Altman comparison method. However, I’m left wondering about the detection of species and the reporting of site richness. It would be interesting to see a table summarising the richness at each reef type according to both methods. I suspect richness is not a monitoring requirement (as you have only used broad Reef Check groupings in the paper) but is of general interest (especially as you say most analysis is done to species level).
An estimate on the additional processing time required for PQ would also be interesting when compared with the chain method.
I would have liked a little more discussion on whether this move to PQ fundamentally supports the objectives of the monitoring (i.e. was the change sufficiently informative to jeopardise a time-series…which I assume it was). Also, the relative pros and cons of both methods would be interesting, i.e. replication vs processing time, detection of rare events (e.g. disease and human litter), transparency etc. I appreciate that much of this is covered in other publications but a brief summary relating to local considerations and objectives would be interesting.

Reviewer 2 ·

Basic reporting

The article is interesting and well written and the subject and research question clearly defined.

The paper as submitted meets the required standard for publication as outlined by PeerJ.

I have made some minor suggestions and noted a few minor typos or errors.

The results section would benefit from a simple summary table of the key findings for the reader, that included the main information. It would be useful to see at a glance which method underestimated or overestimated abundance for each component, within each type of habitat and the conversion required from CPI to PQ.
Line 17: minor typo; should be ‘There are a number ‘
Line 45- not sure being on the ‘frontline’ of the climate change crisis is appropriate wording, you mean that they are highly sensitive?
Line 153: typo should be approach not approached
Line 432- taut not taught, also noted in earlier section but I haven’t recorded line number.
Figure 1- clear and useful
Figure 2 – if retained can be small scale it doesn’t add much of value.
other tables, figures are fine as is the supplementary material.

Appropriate raw data has been made available.

Experimental design

The introduction would have benefited from a sentence or two on the driver behind switching to photoquadrat methods, I assume they are preferred because they also create a long-term photographic method and can be reanalysed? May be more cost-effective, particularly if automated image recognition can be applied in the future?

Other than this caveat, the research question was well defined and set in context. Approaches to integrate time-series data are necessary and assessments of their validity important, so that this study is relevant and meaningful.

The experimental approach provided a rigorous comparison between methods to what appears to be a high technical standard. The analysis and discussion of results were clear and thorough.

The methods were described with sufficient detail to allow replication.

Validity of the findings

The paper provides an assessment of two different methods of monitoring coral reef components. The question is relevant and driven by management requirements. The investigation demonstrated clearly the limitations in comparing methods and the adjustments required to integrate the findings from the different methods. As such it is a clear and useful experiment, the paper fully demonstrates the need for the research, fully articulates the question, addresses this and clearly states the conclusions.

A summary table in the results section would support the reader in rapidly evaluating the results.

The conclusion section was well-written and fully evaluated the findings and discussed the potential underlying drivers of the observed variation between methods. The implications were discussed and the technical detail was presented clearly and without jargon. This research would allow an investigator or organisation to select a method, understand its limitations and support decision making around switching methods. This work was clearly necessary to support on-going monitoring and will have application elsewhere.

All underlying data have been provided.

Additional comments

I found this paper easy to read, interesting and enjoyed reviewing, thank-you.

·

Basic reporting

I found the manuscript clear and unambiguous in describing the potential issues of switching between coral benthic monitoring methodologies without undertaking the necessary comparative study. The intro and background were well thought through and the references cited comprehensive. The layout of the methods, results, and discussion was well done with minimal redundant or over reporting of irrelevant results. Figures were excellent and high quality. Figure 3, in particular, was very well thought through and is an efficient and effective manner of summarizing the results and not typical for previous coral benthic method comparison studies. Table 1 and supplemental tables were also well done.
However, I found table 2 somewhat difficult to understand due in part to the poor description in the results/discussion/caption- why were the N values so different for gorgonians and lower for macroalgae on patch reefs?. Overall, I did not think table 2 added a lot of information to the findings and felt that the discussion of the relative tradeoffs between changing methods and detecting significant change from a managers perspective could be strengthened (please see comments on discussion under Validity of Findings). For table 2, the inclusion of comparative median change values for the 2012-2017 using the 2017 CPI data would provide additional context to the results as presented. I would suggest either removing table 2 (but leaving a the description in the discussion (lines 312-327) of how the PQ method (without necessary conversions) would skew the observed 2012-2017 trends. If the authors wish the table to remain, I would suggest they include a similar column of data for the 2012-2017 CPI changes (median values-1st and 3rd quartiles) for the different reef types/benthic categories.

Experimental design

The study is original primary research relevant to coral reef managers and scientists and within the scope of the Journal. The research questions are well defined and it is articulated how this research fills a current knowledge gap. The study design and methods are well thought out and replicable based on the descriptions and raw data provided. Particularly novel (at least in the coral reef scientific realm) is the robust statistical analysis of differences between the two monitoring methods which is a significant improvement over previous coral monitoring methodological studies.

Validity of the findings

The findings show significant scaling and reef type differences between the two different methods of measuring benthic cover (PQ vs CPI). Previous studies have used simple correlation analysis of results obtained from different methods and the r2 values (often high when spanning a large % cover range) to suggest high compatibility or minimal differences between methods. By utilizing the Bland & Altman (1986) statistical approach, this study provides a much more detailed evaluation of true methodological differences and causes including sample size, scaling, and reef type classification issues. The conclusions area well stated and underscore the potential issues with switching between methodologies without developing necessary scaling and reef type relationships for each benthic class.
The discussion includes a lot of detail about the scaling relationships and potential causes of the differences between the two methods. The significantly higher estimates for gorgonians using the PQ method reinforces previous studies but for many of the other differences (e.g., macroalgae/turf algae) the causes area less clear and some of the relationships appear to contrast with previous studies (e.g. Rogers and Miller, 2001 found relative macroalgae was significantly higher using PQ compared to chain intercept). Observer variance is also minimized in this study as an important factor. However, depending on the procedures used, it may be more important at influencing measurement bias if only one person is doing the lab PQ point counting (as opposed to many observers underwater doing the CPI).
In general, there could be more detail on the practical recommendations of applying corrective measures to historic monitoring datasets to allow for newer PQ methods. In the discussion, lines 319-320 highlight that % cover change is significant from a management perspective when it is 25% or greater and that several of the benthic components exceeded this threshold from methodological bias alone. The implication is that data comparisons between different methods should not be undertaken without the necessary comparative relationships in indicators properly quantified. However, it may not realistic to do comparative studies like this for every long-term monitoring area/reef type/site. So some guidance on applying corrective transformations measured on a subset of sites or from one region to another particularly to coral cover would be a welcomed addition. For example going into some detail on how historic CPI coral cover would be “adjusted using the relationships found in this study (constants, linear equations, etc..) to bring methodologically disparate datasets together based on the 21 sites examined in this study would be quite instructive. More details on the tradeoffs (loss of power to detect change) at the site, reef type, or whole island scale after “adjusting” the historic 1982-2012 CPI datasets to being “PQ comparable” would also be welcomed.

Additional comments

Overall, I found the manuscript to be a solid contribution that exposes the inherent problems of switching between methods and highly recommend its publishing. The two main issues that could strengthen the manuscript are (1) Table 2- suggest ether removing this table or if it stays please add comparative CPI median metrics to allow for a clearer interpretation of the significance of substituting PQ data; (2) expanding the discussion to include some guidance on applying transformations to CPI or PQ datasets in Barbados to minimize the methodological bias quantified in this study.

Additional comments below are some additional suggestions and should not be seen as necessary to address in order to be published.
Why not include correlation plots between the two methods with r2 values (essentially cross plots of supplemental table PQ vs CPI raw data)? This would show a fairly high correlation (r2> 0.8) for most benthic classes across the range of sites but (as this paper goes on to uncover) is somewhat misleading in that these correlations are directly related to the range of % cover values and over look scaling and other issues. Including this would underscore that what on the surface appears to be “highly comparable” methods, is in fact full of significant differences that must be quantified and corrected for to minimize the methodological bias.

What about coral species specific comparison between the two methods? Were the number of coral species or relative proportion significantly different measured by the two methods significantly different?

Cost/benefit comparison between the two methods? Specifically, it would be helpful to know what you found for the relative tradeoffs between the two methods- time underwater vs time in the laboratory/entering data to obtain the results. Even a qualitative statement would be useful additional information for managers in deciding weather or not to switch from PCI to PQ.

---

## Round 0.2 · accepted · Accept

I am entirely satisfied that you have dealt with the excellent reviewer comments satisfactorily.